# Relationship between Hypoxia and Hypercapnia Tolerance and Life Expectancy

**DOI:** 10.3390/ijms25126512

**Published:** 2024-06-13

**Authors:** Pavel P. Tregub, Yulia K. Komleva, Vladimir P. Kulikov, Pavel A. Chekulaev, Oksana F. Tregub, Larisa D. Maltseva, Zaripat Sh. Manasova, Inga A. Popova, Natalia S. Andriutsa, Natalia V. Samburova, Alla B. Salmina, Peter F. Litvitskiy

**Affiliations:** 1Department of Pathophysiology, I.M. Sechenov First Moscow State Medical University, 119991 Moscow, Russia; 2Brain Science Institute, Research Center of Neurology, 125367 Moscow, Russia; yuliakomleva@mail.ru (Y.K.K.);; 3Scientific and Educational Resource Center “Innovative Technologies of Immunophenotyping, Digital Spatial Profiling and Ultrastructural Analysis”, RUDN University, 117198 Moscow, Russia; 4Department of Ultrasound and Functional Diagnostics, Altay State Medical University, 656040 Barnaul, Russia; 5Independent Researcher, 127055 Moscow, Russia

**Keywords:** hypoxia, hypercapnia, aging, gero-protection, naked mole rat, longevity, active longevity, inflamm-aging, nervous system aging

## Abstract

The review discusses the potential relationship between hypoxia resistance and longevity, the influence of carbon dioxide on the mechanisms of aging of the mammalian organism, and intermittent hypercapnic–hypoxic effects on the signaling pathways of aging mechanisms. In the article, we focused on the potential mechanisms of the gero-protective efficacy of carbon dioxide when combined with hypoxia. The review summarizes the possible influence of intermittent hypoxia and hypercapnia on aging processes in the nervous system. We considered the perspective variants of the application of hypercapnic–hypoxic influences for achieving active longevity and the prospects for the possibilities of developing hypercapnic–hypoxic training methods.

## 1. Introduction

Many researchers have long been interested in whether there is a link between hypoxic resistance and longevity and in the possibility of slowing aging by increasing the body’s resistance to damaging effects such as oxygen deprivation and impaired mitochondrial oxidation.

There is much evidence pointing to the potential contribution of hypoxic resistance to both increased longevity and physiologic activity in old age in humans living at high altitudes [1,2,3]. At the same time, there is abundant evidence for increased longevity in vertebrates living underground or in high-altitude and oceanic environments where a low oxygen supply is a major environmental factor [4,5]. For example, the phenomenal longevity of the bowhead whale [6], the naked mole rat [7], and reptiles from the Andean Mountain Range [3] is well-known. Therefore, it is logical to hypothesize that the formation of hypoxic resistance is correlated with opportunities for life extension and may have the gero-protective potential to achieve healthy longevity.

At the same time, it has been proven that creating training hypoxic conditions for the organism increases both the general resistance of the organism [8] and the resistance of individual organs/tissues [9,10,11]. To achieve a protective effect through hypoxic exposures, pre-conditioning regimes (FiO_2_ = 8–12%, 1 to 3 sessions of 30 to 120 min) and intermittent actions (training) (FiO_2_ = 9–16%, 5 to 45 sessions of 48 to 90 min per day) are generally used [11]. There are a lot of data on the preventive and therapeutic efficacy of hypoxic effects in relation to cardiovascular pathology [12,13], trophic disorders [14], and nervous [10,11,15] and respiratory diseases [16]. Further in the manuscript, the terms “intermittent hypoxic exposures/training” will be used to refer to exposure to hypoxia to produce a protective or therapeutic effect by means of respiratory training courses with reduced oxygen in the inhaled air, and the terms “hypoxia/hypoxic injury” will be used to refer to the adverse effects of oxygen deprivation on the organism and/or cells, or when describing the environmental conditions of the organisms’ habitat.

However, there is no accurate evidence or a full assessment of the correlation between the indices of acquired resistance to hypoxic injury and increased longevity of the organism in the literature. In addition, the practical application of the gero-protective potential of intermittent hypoxic effects is difficult due to the need to create long exposure times (1–6 h) and a large number of sessions (at least seven times) [9,12]. Therefore, it is relevant to search for effective techniques to increase the effectiveness of intermittent hypoxic exposures to achieve active longevity and to study the molecular and cellular signaling pathways of potential geronto-protective mechanisms associated with hypoxic resistance.

This review aims to systematize the available knowledge on gero-protective mechanisms that share common signaling pathways with hypoxic resistance, the use of intermittent exercise exposure to hypoxia as a potential gero-protector, the potential role of carbon dioxide in this process, and the identification of likely interrelationships between the different signaling pathways of these mechanisms. These data complement hypotheses about the effect of the mutual potentiation of the protective effects of intermittent exposure to combined hypoxia and hypercapnia [17,18], and the important role of the hypercapnic component in this mechanism [19].

## 2. Hypoxia Resistance in Some “Long-Lived” Species

For many species with relatively long lifespans, the habitat has limited or low oxygen levels [5]. Therefore, various physiological traits have evolved in their bodies to help them adapt to such a harsh ecological environment. Evolutionary adaptations in these species have some fundamental similarities and are aimed at minimizing oxygen deprivation through mechanisms that optimize oxygen uptake or reduce metabolic oxygen demand [4]. Adaptations to hypoxia in these cases include changes in central metabolism, cellular respiration, hemoglobin-mediated oxygen transport, and molecular signaling pathways induced by the transcription factor HIF-1α [20,21,22].

Examples of human adaptations to high-altitude hypoxic conditions that are characterized by comparative longevity include the inhabitants of the Tibetan Plateau [23,24], the Andes, and the Ethiopian Mountains [4], which show strong selection for genes associated with hypoxia adaptation [4,25]. These include, for example, the HIF-associated genes *EPAS1* and *EGLN1*, the *PPARA* gene associated with reduced metabolic oxidation of fatty acids, the hemoglobin gene cluster *HBB/HBG2*, *HFE*, *PKLR*, *CYP17A1*, and *HMOX2*, and additional candidate genes, including *HLa-DQB1/HLA-DPB1*, *ANKH*, *RPaa-384F7.2*, *AC068633.1*, *ZNF532*, and *COL4A4* [4].

Naked mole rats have adapted to the prolonged hypoxia in deep underground burrows [7], while turtles and whales have adapted to conditions of intermittent hypoxia [6,26]. HIF-1 is a key mediator of the hypoxia response pathway, and modulation of this pathway and HIF-1 results in varying degrees of alteration in longevity [27]. In this regard, the hypoxia resistance signaling pathway induced by the HIF-1 factor was presented as a promising target for increasing lifespan in model organisms, such as representatives of Caenorhabditis elegans species [27,28]. This assumption was confirmed in experimental studies. Thus, the loss of VHL-1, responsible for the degradation of HIF-1α, in *C. elegans*, significantly increased their lifespan and increased resistance to polyglutamine and beta-amyloid toxicity [28]. HIF-1 stabilization has also been shown to increase the lifespan of *C. elegans*, but not under conditions of the overexpression of the SKN-1 protein involved in the response to oxidative stress [27]. In a recent study, Tyshkovskiy A., et al. [29] conducted a comprehensive analysis of interventions aimed at prolonging longevity and found that keeping mice under hypoxic conditions (FiO_2_ ≈ 11.8%) for a month induced changes in gene expression similar to the effects of gero-protective interventions. Therefore, it can be considered that an increase in the concentration of HIF-1 and modulation of its signaling pathways can lead to varying degrees of longevity changes in both roundworms and mammals [27]. Interestingly, similar to longevity initiated by hypoxic resistance in model organisms, the establishment of hypoxic conditions in a long-lived species (naked mole rats) induced the activation of longevity-related genes (*HIF-1α*, *HSP-90*, *Hnf-4*, *AhR*, *Ppar*, *Arnt2*, *NcoA1*, and *Rora*) [24,30].

In naked mole rats, adaptation to oxygen deficiency in the atmosphere of underground burrows has become one of the central areas of study of the molecular mechanisms that ensure their longevity [4]. Naked mole rats have a similar pattern of adaptive changes at the genome level, along with physiological activity, which contributes to adaptation to burrowing [31]. Higher expression of genes related to hypoxic resistance, DNA repair, and synthesis of oxygen-transporting globin proteins (*HBA1* and *HBA2*, *NGB*, and *CYGB*) was observed in naked mole rats [32], as well as in highland dwellers (hemoglobin gene cluster *HBB/HBG2*, *HFE*, *PKLR*, *CYP17A1*, and *HMOX2*) [4,25]. According to Lewis et al., oxygen deficiency adaptation in naked mole rats is associated with hypothermia and hypometabolism and is associated with increased longevity [33]. Similarly, mice exposed to a hypercapnic–hypoxic environment show a significant decrease in body temperature and metabolic rate [34,35]. Moreover, some researchers suggest that there is a correlation between high resistance to oxidative stress and hypoxia resistance in naked mole rats [36].

Most notably, there is much evidence of the successful effects on longevity extension in model organisms following intermittent or continuous hypoxic training. For example, subjecting the aquatic invertebrate species *Brachionus manjavacas* to continuous exposure to hypoxia (FiO_2_ ≈ 1.6%) demonstrated a 107% increase in mean lifespan, while reproductive function was doubled in these organisms [37]. In addition, conditions of moderate hypoxia (PO_2_ = 75 mmHg) increased the maximum lifespan of *Drosophila* adults according to Rascón and Harrison [38]. There are also data on the effect on longevity in model organisms through modulation of signaling pathways activated after hypoxic effects. Thus, it was possible to increase the lifespan of *C. elegans* by modulating the HIF-1 pathway [27] and blocking the function of VHL-1 (which negatively regulates HIF-1) [28].

It is important to note that the theory of the relationship between resistance to oxygen deficiency and increased longevity of the organism does not focus only on hypobaric hypoxia in high-altitude conditions, but also takes into account the possibility of forming hypoxic environments in cave lowlands, freezing water bodies, ocean depths, and deep underground [4]. In these cases, organisms in such environments are usually exposed to periodic, intermittent hypoxia, which can be easily modeled using artificial methods. This sets the stage for the potential translation of data on the relationship between acquired hypoxic resistance and increased longevity in the human body.

## 3. The Role of Carbon Dioxide in the Gero-Protective Mechanism in Mammals

Currently, there has been a significant increase in interest in studying the therapeutic efficacy of permissive hypercapnia (a condition caused by excessive amounts of CO_2_ in the inhaled air, with P_CO2_ between 45 mmHg and 100 mmHg), and it has become clear that carbon dioxide in non-toxic doses has a protective effect on the brain in hypoxic/ischemic injury [39,40,41].

Experimental studies on rats have shown that the combination of hypoxia and hypercapnia (hypercapnic hypoxia) in the mode of intermittent training effects gives the maximum increase in resistance to acute hypobaric hypoxia. The results of these studies suggest that hypercapnic hypoxia may be a potential method for prevention, treatment, and rehabilitation, including a means to prolong life and physiological activity in old age. Evidence in favor of this assumption was provided by an important experimental study conducted to evaluate the gero-protective potential of hypercapnic–hypoxic training, which led to a 16% increase in the mean lifespan of mice [42]. In this work, mice of both sexes were exposed to 3-week courses (30 min daily) of intermittent hypercapnic–hypoxic exposure (PO_2_ ≈ 90 mmHg and PCO_2_ ≈ 50 mmHg) every 2 months from puberty until the end of life. The results of this study will be described in more detail in Section 6.

These new data on the protective potential of hypercapnia and hypercapnic hypoxia are optimistic for the authors of this review as supporters of the ideas about the important role of carbon dioxide in the mechanisms of life extension. These ideas are based on the data on the high longevity of animals exposed not only to hypoxia but also to hypercapnia on a regular basis during their lives (whales, freshwater turtles, and naked mole rats) [43,44]. Thus, in long-lived cetaceans and freshwater turtles, in the process of phylogenetic development, the organism has adapted to prolonged stays underwater accompanied by respiratory arrest, in which a state of oxygen deficiency and carbon dioxide excess is formed in the body [22]. In turn, the naked mole rat, living in conditions of poor ventilation underground, forms in the environment a similar deficiency of oxygen and an excess of carbon dioxide [45,46]. It is also noteworthy that the naked male rat was found to be highly tolerant not only to anoxia/hypoxia (several hours at FiO_2_ = 3% and several days or weeks at FiO_2_ = 8%) but also to extreme hypercapnia (FiCO_2_ ≥ 10%) [46,47].

Another interesting fact is the change in the expression of carboanhydrase genes in the naked mole rat under hypoxic conditions (FiO_2_ = 8%) (decreased expression of Ca1–3 in the liver and Ca4 in the brain; increased expression of Ca12–14), which may be related to a mechanism to control CO_2_ and bicarbonate concentrations, since these carboanhydrases catalyze the reversible hydration of CO_2_ and are involved in maintaining cellular pH [30]. This suggests a genomic relationship between the longevity of the naked mole rat and its resistance to hypercapnia. It is also noteworthy that the change in the expression of the above genes under the influence of oxygen deficiency occurs under normocapnic conditions, which indicates the preventive activation of signaling pathways to increase the resistance of the organism to excess CO_2_ (which often accompanies hypoxic damage) and the presence of common regulatory mechanisms between hypercapnic and hypoxic resistance.

It should be noted that the role of carbon dioxide in the gero-protective effect of intermittent hypoxic exposures can be implemented due to its influence on molecular and cellular mechanisms of adaptation to harmful stimuli: stimulation of angiogenesis, enhancement of the antioxidant activity of cellular enzymes, modulation of apoptosis, increase in proliferative activity, reprogramming of mitochondrial metabolism, and stimulation of chaperones [48].

## 4. Signaling Pathways of the Gero-Protective Potential of Intermittent Hypercapnic–Hypoxic Exposures

According to recent data, increased CO_2_ (PCO_2_ ≈ 50 mmHg) levels under intermittent hypercapnic–hypoxic exposure have a positive effect on a number of key gero-protective molecular cellular mechanisms [48]. Among these mechanisms, a number of researchers highlight the active activation of angiogenesis [49,50,51]. In this case, a 2-week course of daily 20-min exposure to hypercapnic hypoxia (PO_2_ ≈ 90 mmHg and PCO_2_ ≈ 50 mmHg) activates neo-angiogenesis signaling pathways mediated by increased levels of VEGF in the body [52]. At the same time, it is known that impaired angiogenesis is associated with cognitive decline in the elderly, and exercise increases the concentration of specific markers in this category of people, including VEGF [53]. Interestingly, this factor is also a target for the therapy of age-related diseases: neovascular age-related macular degeneration [54] and poor regeneration of bone and cartilage apparatus [55].

Another potential mechanism of the gero-protective effect of carbon dioxide may be its antioxidant efficiency. Thus, a moderate level of CO_2_ stimulates the antioxidant activity of intracellular enzymes [56,57]. This mechanism is probably related to the activation of glutathione peroxidase [56] and superoxide dismutase [57,58]. In particular, Barth A. et al. [56] showed that after 60 min of hypoxia (FiO_2_ = 8%) with hypercapnia (FiCO_2_ = 10%) in piglets, there was a more pronounced increase in the amount of reduced glutathione and a decrease in the amount of oxidized glutathione compared to animals exposed to conditions of intrauterine hypoxia. At the same time, Goss S.P.A. et al. [58] demonstrate that bicarbonate (a CO_2_-dependent component of the body buffer system) enhances the peroxidase activity of superoxide dismutase, presumably through the formation of a carbonate radical anion. In addition, CO_2_ may affect the stability of the iron–transferrin complex [57], preventing the participation of iron ions in the initiation of free-radical reactions [57,59]. In addition, carbon dioxide can neutralize reactive oxygen/nitrogen species by combining with peroxynitrite and converting to nitrocarbonate, which forms carboxy and nitroxide anions when combined with water [60,61]. Importantly, moderate 4 h exposure to hypoxia (PO_2_ ≈ 30 mmHg) and hypercapnic hypoxia (PO_2_ ≈ 30 mmHg and PCO_2_ ≈ 7.5 mmHg) in *Litopenaeus vannamei* promotes the activation of the antioxidant system by increasing the expression of Mn-superoxide dismutase, glutathione peroxidase, and peptide-methionine (R)-S-oxidoreductase genes [62]. Such data serve as important evidence for the potential role of the hypercapnic component in the mechanism of defense against aging, which is largely due to oxidative stress and free radical reactions [63,64]. It should be noted that in the naked mole rats (resistant to hypoxia and hypercapnia), an effective antioxidant status is also considered by a number of researchers to be a likely mechanism for the observed longevity [36,65].

Modulation of apoptosis [66] and enhancement of proliferative cellular activity [67] can be considered as other important gero-protective effects of hypercapnia, including its combination with hypoxic exposure, shown in in vitro models on cultures of rat astrocytes and neurons, as well as in vivo in the peri-stroke area of the cerebral cortex. Under certain conditions, this mechanism may be the cause of malignant transformation of cells; however, in combination with the effective immune defense of the organism and reparative functions of the epigenomic apparatus of the cell, it may be a component of the strategy of protection against aging, especially of the brain and myocardium [68,69].

Among the potential gero-protective mechanisms of hypercapnia exposure combined with hypoxia is its effect on signaling pathways activated by heat shock proteins, which are cellular chaperones [52]. The increased gene expression of heat shock proteins is a universal cellular response to damage, and their chaperone activity provides cyto-protection during stress [70]. At the same time, compensation of intracellular stress and the associated inter-organellar response of the cytosol, mitochondria, and endoplasmic reticulum is recognized as an important link in the implementation of cellular homeostasis in the process of increasing the chronological age of the organism [71,72,73].

In connection with the above data, the effect of hypoxia and hypercapnia can be considered from the point of view of factors that, in moderate doses, have a modulating effect on the signaling mechanisms controlled by the vitagenes system (heat shock proteins and the thyredoxin and sirtuin protein system), which plays an essential role in protecting cells from stress and prolonging the organism’s life [74,75]. It is possible that the nature of this effect is mediated by epigenetic mechanisms through the alteration of the expression of genes encoding protective factors, for example, through the influence of the transcription factor NF-κB, which increases after exposure to hypercapnia [76], and HIF-1α/-2α, which increases during hypoxic exposure [77,78]. It is also interesting that after the naked shrew is subjected to hypoxic exposure, the expression of genes of the S100 protein family (*S100A8* and *S100A9*), which enhance the effects of NF-κB, is also increased in the transcriptome of liver, kidney, and brain cells, while the expression of *S100B* and *S100P* genes, which cause inhibition of apoptosis, is decreased [30]. This again suggests the likelihood of common regulatory mechanisms between hypercapnic and hypoxic resistance. It should be noted that most of the molecular and cellular mechanisms involved in the formation of potential gero-protective mechanisms are influenced by both excess carbon dioxide and oxygen deficiency (Figure 1).

## 5. Influence of the Protective Effects of Hypercapnia and Hypoxia on the Mechanisms of Nervous System Aging

The aging process of the individual tissues of the whole organism is reflected differently in physiological activity in old age. Of particular interest in this regard are tissues that are largely composed of cells that do not proliferate during the mature period of ontogenesis (nervous and cardiac), as the functions of many vital organs depend on the reliability of their cyto-protective and reparative mechanisms [79,80]. Therefore, much attention from neurobiologists has been paid to the pathogenesis of the mechanism of nervous system aging in age-associated neurological diseases, such as Alzheimer-type neurodegeneration [81].

Neurodegeneration and aging are intricately linked to how cells, the brain, and the body as a whole respond to oxygen deprivation [82]. The main pathological processes involved in neurodegeneration, such as neuroinflammation, oxidative stress, and mitochondrial dysfunction, also play significant roles in the aging process. In fact, inflammation, oxidative stress, and mitochondrial issues tend to worsen as we age, contributing significantly to the aging process itself. During the aging process, there is a notable rise in systemic low-grade inflammation, commonly referred to as inflamm-aging, along with immunosenescence [83,84].

The state of hypoxia has the dual ability to either trigger or suppress these processes, including inflammation, oxidative stress, and mitochondrial dysfunction. Molecular signaling pathways controlled by the HIF transcription factor are crucial in mediating these effects. Given the overlap in cellular mechanisms involved in aging, neurodegeneration, and reactions to oxygen deficiency, utilizing intermittent hypoxia conditioning to address oxidative stress, mitochondrial dysfunction, and neuroinflammation could offer a promising treatment avenue for neurodegenerative conditions [82].

However, there are diverse viewpoints regarding the influence of hypoxic exposure on brain aging. Numerous studies indicate that hypoxic episodes can have both neuroprotective and neurotoxic effects depending on factors such as severity, frequency, and duration of exposure [85,86,87]. Acute sustained hypoxia, lasting less than 4 h, appears to prime cells for subsequent damage by activating protective pathways within neurons [88]. Conversely, chronic intermittent hypoxia, occurring nightly during sleep, is linked to aggravated neurodegeneration in various brain regions, potentially affecting functions such as cognition, motor skills, and homeostasis [87]. Elevated oxidative stress and inflammation, both hallmark features of neurodegenerative diseases, are observed during sleep apnea [89]. Furthermore, high-altitude hypoxia induces brain function alterations reminiscent of those in normal and accelerated aging (e.g., Alzheimer’s disease), jeopardizing cognitive and executive functions. Recent research has confirmed high-altitude hypoxia as a model for assessing age-related brain activity disturbances [90]. Data suggest that intense, episodic hypoxia is associated with shortened telomere length, another marker of aging, in leukocytes of patients with obstructive sleep apnea [91].

Another confirmation of the negative role of hypoxic damage can be seen in the development of cognitive impairments after recovery from COVID-19. Hypoxic damage, as one of the main clinical manifestations of severe SARS-CoV-2 infection, directly or indirectly contributes to premature neuronal aging, neuroinflammation, and neurodegeneration by altering the expression of various genes responsible for cell survival [92].

These research findings indicate that chronic intermittent hypoxia leads to heightened circulating oxidative stress and inflammation. Furthermore, brain regions associated with early (but not late) stages of Alzheimer’s disease and Parkinson’s disease exhibited oxidative stress and inflammatory profiles, aligning with observations in preclinical populations of neurodegenerative diseases. These results suggest that mild chronic intermittent hypoxia induces key features characteristic of the early stages of neurodegenerative diseases and may serve as an effective model for studying mechanisms contributing to oxidative stress and inflammation in these brain regions [87].

However, a significant portion of research demonstrates that the influence of hypercapnia and hypoxia can exert cyto-protective effects on neuronal survival and function. These beneficial effects are mediated by various molecular pathways that serve to enhance cell resilience and mitigate damage in response to stressful stimuli. Key molecular pathways involved in cyto-protection include the activation of hypoxia-inducible factors, which regulate the expression of genes involved in oxygen homeostasis, energy metabolism, and cell survival. Additionally, hypercapnia and hypoxic pre-conditioning lead to increased levels of antioxidant enzymes such as superoxide dismutase and catalase, which counteract oxidative stress and prevent cellular damage. Moreover, these environmental stressors can modulate apoptotic pathways, thereby reducing neuronal apoptosis and promoting cell survival under challenging conditions [48].

In general, the cyto-protective effects of intermittent exposure to hypercapnia and hypoxia hold potential therapeutic value for various neurological disorders, including neurodegenerative diseases. Further elucidation of the molecular mechanisms and optimization of therapeutic strategies may pave the way for new interventions aimed at enhancing the health and resilience of neurons in the face of aging and pathological damage.

A moderate reduction in mitochondrial respiration, leading to decreased oxygen utilization, has been shown to extend the viability of cultured cells and increase lifespan in mice [93], consistent with the proposed inverse relationship between metabolic rate and lifespan [94]. Apart from the HIF system, aging alters oxygen sensitivity and the ventilatory response to oxygen deficiency. Aging is associated with decreased respiratory capacity due to changes in the function of peripheral and central chemoreceptors and reduced ventilatory pump power due to the decreased strength of respiratory muscles and chest wall compliance [82]. Furthermore, the age-related increase in HIF-1α content and expression of HIF target genes, such as VEGF and iNOS, caused by hypoxia-induced decreases in cellular oxygen levels, decreases with age [95].

In addition to the age-related decline in HIF signaling and variations in the impact of hypoxia on brain function, other observations suggest an altered, sometimes weakened, or delayed ability to adapt to oxygen deficiency in elderly individuals [96]. Adaptation to moderate intermittent hypoxia training may enhance cerebral oxygenation and hypoxia-induced cerebrovasodilatation while improving short-term memory and attention in elderly patients with MCI [97]. At the same time, it has been shown that carbon dioxide acts as an important regulator of cerebral hemodynamics when the oxygen level in arterial blood decreases [98], which may also improve brain function by increasing blood flow during exercise with moderate intermittent hypoxia and hypercapnia.

It is well known that sleep breathing disorders are much more common in the elderly. In this case, central sleep apnea is characterized by chronic hyperventilation and hypocapnia, and attacks of respiratory failure with hypercapnia stimulate hyperventilation and hypocapnia in the interictal period in bronchial asthma and nocturnal apnea [99,100]. It should also be noted that studies have shown a correlation of reduced cerebrovascular reactivity to CO_2_ with the occurrence of central sleep apnea in patients with congestive heart failure [101], as well as a relationship with obstructive sleep apnea [102]. In this regard, it is important to note that the intermittent training effects of hypercapnia lead to the normalization of the sensitivity of carotid body chemoreceptors to CO_2_, which reduces hypertension and restores autoregulation of cerebral circulation [103]. There is also evidence that cerebral blood flow decreases with age due to a decrease in PaCO_2_, and exposure to hypercapnia in the elderly can eliminate the age-related decrease in cerebral blood flow by 50% [104]. These data, in turn, are additional prerequisites for achieving positive effects of intermittent hypercapnic–hypoxic effects in normalizing cerebral blood flow in the elderly.

Respiratory training with intermittent hypercapnic hypoxia in patients with arterial hypertension has also been shown to stabilize blood pressure in hypertension by compensating for the effects of hypocapnia occurring under conditions of chronic stress [103]. This mechanism is achieved by normalizing the sensitivity of the carotid body chemoreceptors to CO_2_ and reducing the sympathetic influences on the heart and vessels. In addition, hypercapnic training restores cerebrovascular autoregulation disturbed in conditions of arterial hypertension, which also has a positive gero-protective effect on the central nervous system [103].

There is evidence that severe hypoxia (PO_2_ = 59 mmHg) induces anxious behavior in rats [105], but intermittent hypoxia (FiO_2_ = 9–16%) improves cognition and reduces anxiety in a mouse model of Alzheimer’s disease [106]. In addition, moderate intermittent hypoxia can alleviate psychological distress and depression [107], which also often accompany aging. In the experimental study, intermittent hypoxia significantly reduced the number of Aβ plaques in the hippocampus of Alzheimer’s disease mice. The recovery of cognition correlated with the increase in proliferation and differentiation of neural stem cells, which suggests that intermittent hypoxia may exert its beneficial effect by restoring neurogenesis. It was assumed that the mechanism underlying hypoxia-stimulated neurogenesis is related to the restoration of BDNF signaling [108]. There is also evidence that 50-min intervals of intermittent hypoxia can potentially increase brain oxygenation [109]. The beneficial effects of intermittent exposure to hypercapnia and hypoxia on the molecular cellular pathways of cyto- and gero-protection may serve as promising predictors for approaches to the development of gero-protective strategies aimed at slowing brain aging processes.

To summarize, regarding the interaction between hypoxia and aging, the following key points can be highlighted: an effect that causes oxygen deficiency in the brain depends on the type, severity, duration, and frequency of hypoxia exposure. While severe oxygen deficiency is detrimental to the brain and is involved in the pathogenesis of neurodegenerative diseases, brain adaptation to moderate hypoxia can be used for neuroprotection, and training exposure to intermittent hypoxia under hypoxic conditions is a potential strategy for the treatment of neurodegenerative diseases.

## 6. Perspectives on Translational Research

In one of our experimental studies mentioned above [42], in addition to assessing the change in average lifespan, the effect of intermittent hypercapnic hypoxia on the main criteria of its quality (reproductive function, muscle strength, and behavioral activity) was also determined. The results of the study showed that the average life expectancy of mice exposed to intermittent hypercapnic hypoxia increased by 16%. When interpreted to the average human lifespan of approximately 70 years, this is equivalent to 11 years [110]. When the above data are compared with the results of similar experimental studies, it can be seen that this is a significant achievement. For example, when evaluating the effectiveness of genetically induced activation of the telomerase enzyme, the average life expectancy of mice was increased by 18% [111], and when rapamycin (an immunosuppressive drug and an mTOR kinase inhibitor) was added to the diet of older mice, life expectancy was increased by 13% [106]. It is important to note that the drugs used in the above studies are characterized by side effects, particularly the risk of tumors [112]. At the same time, exposure to intermittent hypercapnic hypoxia showed a tendency toward oncoprotection against the development of tumors characteristic of mice [113], which favorably distinguishes hypercapnic–hypoxic exposure from the above-mentioned drugs. This fact is also consistent with the fact that in long-lived species (e.g., naked mole rats and Greenland whales), resistance to oncopathology is associated with increased longevity [114,115]. 

In addition, experiments show that exposure to intermittent hypercapnic hypoxia prolongs the reproductive age of mice [42]. This, together with the positive effects on muscle strength and physical fatigue in old age, is an important contribution to support the hypothesis that intermittent hypercapnic hypoxia has a protective potential in old age, as these indicators are one of the key components of a long and healthy life [116]. Meanwhile, regular exposure to intermittent hypercapnia–hypoxia has been shown to improve stress tolerance and cognitive performance in mice and to activate exploratory behavior in old age. These improvements can also be considered as indicators of healthy longevity [117].

It should be emphasized that the nature of aging is very multifaceted and depends on many internal and external factors acting on cells, tissues, and the organism as a whole. Therefore, the effects of potential gero-protective drugs or environmental conditions (e.g., exposure to hypoxia and/or hypercapnia) may involve only some components of the mechanism of the aging process. In addition, it is important to extrapolate the results of studies obtained in animals to humans with caution, as comparisons between human and animal physiological aging processes are complex and often unreliable.

Thus, it can be suggested that exposure to intermittent hypercapnic hypoxia may be a promising method of prolonging life and physiological activity in old age. This shows the need for further studies to evaluate the gero-protective potential of hypercapnic hypoxia with assessments of biological age markers (humoral, reproductive, biochemical, and functional) using experiments, with healthy volunteers under the condition of beginning training in old age.

## Figures and Tables

**Figure 1 ijms-25-06512-f001:**
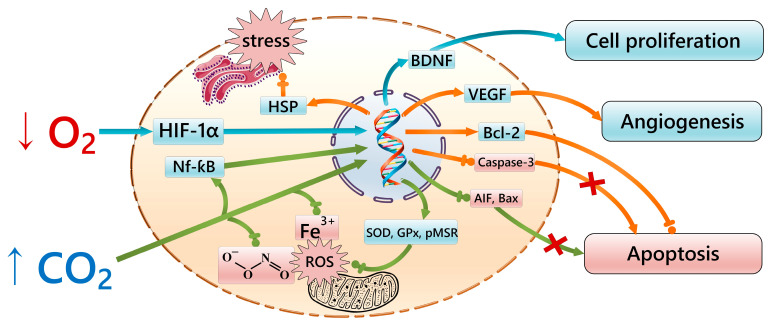
Signaling pathways of the gero-protective potential of intermittent hypercapnic–hypoxic exposures. Blue lines indicate the effects of hypoxia; green lines indicate the effects of hypercapnia; orange lines indicate combined effects. HIF-1α—hypoxia-inducible factor 1-alpha; SOD—Superoxide Dismutase; GPx—Glutathione Peroxidase; pMSR—Peptide Methionine(R)-S-Oxide Reductase; HSP—heat shock protein; NF-κB—Nuclear factor kappa-light-chain-enhancer of activated B cells; Bax—Bcl-2-associated X protein; Bcl-2—B-cell lymphoma 2; AIF—Apoptosis-inducing factor; VEGF—Vascular endothelial growth factor; BDNF—brain-derived neurotrophic factor; ROS—reactive oxygen stress.

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
