# Peer review of "Relationship between Hypoxia and Hypercapnia Tolerance and Life Expectancy"

_ijms, 2024, doi:10.3390/ijms25126512_

Round 1

Reviewer 1 Report

Comments and Suggestions for Authors

The manuscript, "Relationship between Hypoxia and Hypercapnia Tolerance and Life Expectancy," explores the relationship between hypoxia, carbon dioxide influence, and mammalian aging mechanisms. While the selected topic is highly relevant to hypoxia research, the manuscript requires further improvement before publication. It provides a superficial discussion with ambiguous sections and assumptions that are still under investigation.

One significant issue that needs to be addressed is the inconsistent use of the term "hypoxia." It is sometimes used ambiguously, such as in phrases like "hypoxic effect" or "hypoxic conditions" when referring to hypoxic preconditioning. These terms represent distinct phenomena and should be clarified to avoid any potential confusion.

Furthermore, the manuscript lacks coherence and structure. To enhance its quality, authors should delve deeper into the topic by providing a more comprehensive description of the original articles consulted, including details on experimental conditions (e.g., hypoxia or hypercapnia duration, oxygen or carbon dioxide concentrations, and experimental models). Relying more on primary sources than literature reviews is crucial to strengthening the argumentation and ensuring the accuracy of the information presented.

In addition to these points, authors should consider the following:

  1. Exercise caution when extrapolating findings from animal studies to humans, as the translation of results to human physiology and lifespan extension is complex and often unreliable.
  2. Avoid oversimplifying the multifaceted nature of aging by attributing longevity solely to specific environmental conditions, such as hypercapnic hypoxia. 

To improve the manuscript, the authors should address these issues and carefully consider the attached revised version, which contains specific comments on particular sections.

Author Response

The author sincerely thanks the reviewer 1 for the careful review of our manuscript and expert assistance. We hope that we have been able to answer the questions and our manuscript has been improved after revising it according to the recommendations provided.

Below are the answers to the questions and recommendations that were also added to the manuscript:

«One significant issue that needs to be addressed is the inconsistent use of the term "hypoxia." It is sometimes used ambiguously, such as in phrases like "hypoxic effect" or "hypoxic conditions" when referring to hypoxic preconditioning. These terms represent distinct phenomena and should be clarified to avoid any potential confusion.»

We have revised the manuscript to correct this observation and in the Introduction section (3rd paragraph) made clarifications in terminology for the reader.

«Furthermore, the manuscript lacks coherence and structure. To enhance its quality, authors should delve deeper into the topic by providing a more comprehensive description of the original articles consulted, including details on experimental conditions (e.g., hypoxia or hypercapnia duration, oxygen or carbon dioxide concentrations, and experimental models). Relying more on primary sources than literature reviews is crucial to strengthening the argumentation and ensuring the accuracy of the information presented.»

- Thank you very much for this valuable reference. We have endeavored to supplement the manuscript with detailed information from primary sources, and we have also contributed some additional data in terms of discussing the results of the genomic analysis of the naked mole rat and comparing them with data from other sources (Section 3, paragraphs 2 and 4; Section 4, paragraph 5).

«Exercise caution when extrapolating findings from animal studies to humans, as the translation of results to human physiology and lifespan extension is complex and often unreliable. Avoid oversimplifying the multifaceted nature of aging by attributing longevity solely to specific environmental conditions, such as hypercapnic hypoxia. »

We fully agree with you on these statements. We have therefore added a relevant fragment to the manuscript (Section 6, paragraph 3), and we have clarify terms we used in the manuscript.

To improve the manuscript, the authors should address these issues and carefully consider the attached revised version, which contains specific comments on particular sections.

- We have made appropriate corrections and made some comments in the review (attached to the response).

Reviewer 2 Report

Comments and Suggestions for Authors The manuscript by Pavel P. Tregub and colleagues presents evidence for hypoxia resistance and longevity. The authors acknowledge that further studies based on the current literature are needed and suggest that hypercapnic hypoxia may be a promising approach to prolong life and preserve physiological functions in old age. The review is well structured and draws on various publications on this topic. However, some points require further discussion: 1. The prevalence of sleep-disordered breathing is higher in older adults than in younger people. The increased susceptibility to unstable ventilatory control in older people may contribute to a higher incidence of central apnea. Reduced cerebral vascular response to CO2 may exacerbate over- and under-ventilation during sleep. 2. In addition, cerebral vasodilation in response to hypercapnia has been observed to be lower in the elderly than in younger people. 3. Sleep apnea is a common comorbidity in neurodegenerative diseases such as Alzheimer's disease (AD) and Parkinson's disease (PD). 4. The problem of hypoxia-induced alkalosis and acidosis needs to be addressed. Alkalosis is known to decrease sympathetic tone, attenuate hypoxic pulmonary vasoconstriction and cerebral vasodilation, and increase hemoglobin-oxygen affinity. 5. Hypoxemia leads to metabolic and hypercapnic acidosis, accompanied by significant lactate production, resulting in a drop in pH below 6.8.
I recommend approaching the work not only from the point of view of the advantages but also of the possible negative effects.
Comments on the Quality of English Language In general, the English seems to be correct, but there are problems with the construction of some sentences.

Author Response

We thank the reviewer 2  for the evaluation of our work and formulated questions to the manuscript

Below are the answers to the questions and recommendations, which were also added to the manuscript:

  1. The prevalence of sleep-disordered breathing is higher in older adults than in younger people. The increased susceptibility to unstable ventilatory control in older people may contribute to a higher incidence of central apnea. Reduced cerebral vascular response to CO2 may exacerbate over- and under-ventilation during sleep.

Sleep breathing disorders are much more common in the elderly. But importantly, central sleep apnea is characterized by chronic hyperventilation and hypocapnia. Attacks of respiratory failure with hypercapnia stimulate hyperventilation and hypocapnia in the interictal period in bronchial asthma and nocturnal apnea [Fried R. The hyperventilation syndrome-research and clinical treatment. J Neurol Neurosurg Psychiatry. 1988 Dec;51(12):1600-1. doi: 10.1136/jnnp.51.12.1600-b; Nardi A.E., Valenca A.M., Lopes I., Nascimento M.A., Mezzasalma W.A. Zin. Clinical features of panic patients sensitive to hyperventilation or breath-holding methods for inducing panic attacks. Braz J Med Biol Res. 2004 Feb;37(2):251-7. doi: 10.1590/s0100-879x2004000200200013; Grishin OV. Psychogenic dyspnea and hyperventilation syndrome: monograph. Novosibirsk: Manuscript, 2012.-224 p.In Russian].

It should be noted that studies have shown a correlation between reduced cerebrovascular reactivity to CO2 and the occurrence of central sleep apnea in patients with congestive heart failure [Xie A, Skatrud JB, Khayat R, Dempsey JA, Morgan B, Russell D. Cerebrovascular response to carbon dioxide in patientswith congestive heart failure. Am J Respir Crit Care Med. 2005;172 (3):371-8. doi:10.1164/rccm.200406-807OC], and the association with obstructive sleep apnea [Burgess KR, Fan JL, Peebles KC, Thomas KN, Lucas S, Lucas R et al. Exacerbation of obstructive sleep apnea by oral indomethacin. Chest. 2010;137(3):707-10. doi:10.1378/chest. 09-1329]. In this regard, it is important to note that intermittent training effects of hypercapnia leads to normalization of sensitivity of carotid body chemoreceptors to CO2, which reduces high blood pressure and restores autoregulation of cerebrovascular circulation [Kulikov V.P., Kuznetsova D.V., Zarya A.N.. ROLE OF CEREBROVASCULAR AND CARDIOVASCULAR CO2-REACTIVITY IN THE PATHOGENESIS OF ARTERIAL HYPERTENSION. “Arterial'naya Gipertenziya (”Arterial Hypertension"). 2017;23(5):433-446. https://doi.org/10.18705/1607-419X-2017-23-5-433-446]

In addition, we insist on the position that intermittent exposure to hypercapnic hypoxia forms positive adaptive changes in the organism due to moderate load (sessions of 30 minutes, once a day). In contrast, nocturnal apnea leads to frequent, prolonged and uncontrolled episodes of breath-holding, which lead to hypercapnic hypoxia, which has negative consequences, including on the state of the cardiovascular system [Shao C, Wang H, He Y, Yu B, Zhao H. Clinical phenotype of obstructive sleep apnea in older adults: a hospital-based retrospective study in China. Ir J Med Sci. 2023 Oct;192(5):2305-2312. doi: 10.1007/s11845-023-03290-0.]. Therefore, we do not identify the effects of intermittent exposures to hypercapnic hypoxia with components of the pathogenesis of nocturnal apnea.

  1. In addition, cerebral vasodilation in response to hypercapnia has been observed to be lower in the elderly than in younger people.

Yes, we support this statement. Cerebrovascular CO2 reactivity, which reflects the ability to dilate cerebral resistive vessels, decreases with age.. Age-related decline in cerebrovascular reactivity has significance in the formation of arterial hypertension (Lipsitz LA, Mukai S, Hammer J et al. Dynamic regulation of middle cerebral artery blood flow veloci- ty in aging and hypertension. Stroke 2000; 31: 1897-903.], nocturnal apnea [Xie A, Skatrud JB, Khayat R, Dempsey JA, Morgan B, Russell D. Cerebrovascular response to carbon dioxide in patientswith congestive heart failure. Am J Respir Crit Care Med. 2005;172 (3):371-8. doi:10.1164/rccm.200406-807OC; Burgess KR, Fan JL, Peebles KC, Thomas KN, Lucas S, Lucas R et al. Exacerbation of obstructive sleep apnea by oral indomethacin. Chest. 2010;137(3):707-10. doi:10.1378/chest. 09-1329], Alzheimer's disease [Glodzik L., Randall C., Rusinek H., de Leon M.J.. Affiliations expand Cerebrovascular reactivity to carbon dioxide in Alzheimer's disease. J Alzheimers Dis. 2013;35(3):427-40.  doi: 10.3233/JAD-122011] and likely other diseases of the elderly. However, elderly normotensive patients and patients receiving antihypertensive therapy retain the ability to autoregulate cerebral blood flow  [Kulikov V.P., Kuznetsova D.V., Zarya A.N. ROLE OF CEREBROVASCULAR AND CARDIOVASCULAR CO2-REACTIVITY IN THE PATHOGENESIS OF ARTERIAL HYPERTENSION. "Arterial’naya Gipertenziya" ("Arterial Hypertension"). 2017;23(5):433-446. https://doi.org/10.18705/1607-419X-2017-23-5-433-446]. In this regard, it is important that training with hypercapnic hypoxia has an antihypertensive effect and normalizes autoregulation of cerebral circulation.

There is also evidence that during aging decreases in cerebral blood flow due to a decrease in РаСО2, and exposure to hypercapnia in the elderly can eliminate the age-related decrease in cerebral blood flow by 50%. [Flück D, Braz ID, Keiser S, Hüppin F, Haider T, Hilty MP, Fisher JP, Lundby C. Age, aerobic fitness, and cerebral perfusion during exercise: role of carbon dioxide. Am J Physiol Heart Circ Physiol. 2014 Aug 15;307(4):H515-23. doi: 10.1152/ajpheart.00177.2014.].

All these data, in turn, are an additional for the beneficial effects of intermittent hypercapnic-hypoxic exposures in normalizing cerebral blood flow in the elderly. (The above excerpts have been added to the manuscript)

  1. Sleep apnea is a common comorbidity in neurodegenerative diseases such as Alzheimer's disease (AD) and Parkinson's disease (PD).

- The answer to this point is contained in the previous comments.

  1. The problem of hypoxia-induced alkalosis and acidosis needs to be addressed. Alkalosis is known to decrease sympathetic tone, attenuate hypoxic pulmonary vasoconstriction and cerebral vasodilation, and increase hemoglobin-oxygen affinity.

We agree that the pronounced alkalosis that hypoxia induces in the body forms negative effects in  cerebral blood supply and tissue oxygen supply (due to increased affinity of hemoglobin for oxygen). The development of alkalosis or acidosis in hypoxia is determined by the type of hypoxia and the associated РаСО2 measurement. The mechanism for the development of alkalosis in hypoxia is well known from numerous classical works on high altitude hypobaric hypoxia. It is based on hyperventilation and hypocapnia. The same mechanism is characteristic of exogenous normobaric hypoxia during inhalation of an oxygen-poor gas mixture in an open circuit.

However, in exogenous hypoxia with confined breathing, hypoxia is combined with hypercapnia, resulting in acidosis. Acidosis accompanies other types of hypoxia with hypercapnia. We, in turn, used hypercapnic hypoxia, which forms in the body a state of moderate acidosis, having opposite effects on pulmonary and cerebral hemodynamics and oxygen transport function of hemoglobin. The development of acidosis during such exercise directly depends on the concentration of CO2 in the inhaled gas mixture and the duration of exercise. A detailed comparison of indicators of the acid-base state of arterial blood during hypoxic, hypercapnic or hypercapnic exposures was performed in our other work, which is beyond the scope of the current review [V. P. Kulikov,1 Yu. G. Motin,2 P. P. Tregub,1 P. D. Kovzelev,1K. A. Shoshin,1 E. K. Zinchenko,1 A. E. Chernetsky1. COMBINED HYPERCAPNIA AND HYPOXIA LEAD TO THE ACIDOSIS AND INCREASE THE AMOUNT HIF-1a IN RAT HIPPOCAMPUS. RUSSIAN JOURNAL OF PHYSIOLOGY. V. 104 N 11 P. 1347-1355. 2018]

  1. Hypoxemia leads to metabolic and hypercapnic acidosis, accompanied by significant lactate production, resulting in a drop in pH below 6.8.

The answer to this point is contained in the previous comment.

I recommend approaching the work not only from the point of view of the advantages but also of the possible negative effects.

We fully agree with you in this statement. Therefore, we would like to point out that the relevant statements about the necessity to create training effects in moderate doses and regimes are indicated in the text of the manuscript in Section 4 (Paragraph 5) and Section 5 (Paragraphs 2 - 6, 10, 12 and especially 13).

Round 2

Reviewer 1 Report

Comments and Suggestions for Authors

The presentation of the manuscript improved substantially after the authors' revision, so it is currently almost ready to be published. However, the authors still need to review their use of abbreviations and correct some details regarding writing in English.

Comments on the Quality of English Language

The authors should revise the meaning of the verb realize; it is not a synonym for perform or do. In addition, the authors should thoroughly review the text to avoid unnecessary words, for example, "may be can be".

Author Response

Dear reviewer!

Thank you for your valuable comments!

Corrections of the scientific language have been made to the manuscript.

Reviewer 2 Report

Comments and Suggestions for Authors After careful reading of the considerations presented by the authors, I believe that my demands in this regard have been met. The manuscript has been completely revised in its present form and appears very convincing.

Author Response

Dear reviewer!

Thank you for your valuable comments!